# Slaughter of Pregnant Cattle at an Austrian Abattoir: Prevalence and Gestational Age

**DOI:** 10.3390/ani11082474

**Published:** 2021-08-23

**Authors:** Ignaz Zitterer, Peter Paulsen

**Affiliations:** 1Department of Health, Youth and Family, Veterinary Services, Municipality of the Provincial Capital Klagenfurt am Wörthersee, Schlachthofstraße 7, 9010 Klagenfurt am Wörthersee, Austria; Ignaz.zitterer@klagenfurt.at; 2Unit of Food Hygiene and Technology, Institute of Food Safety, Food Technology and Veterinary Public Health, University of Veterinary Medicine Vienna, Veterinärplatz 1, 1210 Vienna, Austria

**Keywords:** Austria, cow, heifer, gestation stage, pregnancy, Simmental, slaughter

## Abstract

**Simple Summary:**

Cattle constitute a major part of the livestock in Austria. Dairy cows are sent to slaughter at the end of their production cycle, whereas heifers are admitted to slaughter either after a fattening period or because of reproductive disorders. In several countries, evidence has been presented that pregnant female cattle are admitted to slaughter, with implications for animal welfare and meat quality. Until today, no data are available on the frequency of pregnant cattle slaughtered in Austria. Over a one-year period, we examined 1633 female cattle in one abattoir, and pregnancy was detected in 104 animals (6.4%). Sixteen cattle were in the last third of gestation. Percentages of pregnant cattle sent to slaughter were higher in beef and dual-purpose breeds than in dairy cattle, but this was not statistically significant. Measures to minimize the number of pregnant cattle sent to slaughter should be implemented at farm-level.

**Abstract:**

The slaughter of pregnant cattle raises ethical–moral questions with regard to animal welfare, but also concerns of consumers because of higher levels of sex steroids in the meat from pregnant cattle. Since no data on the slaughter of pregnant cattle in Austria were available, we examined uteri of slaughtered female cattle in one Austrian mid-size abattoir. Sample size was calculated for an assumed prevalence of 2.5% (±1%; 95% confidence interval) of cows or heifers slaughtered in the last trimester of pregnancy and amounted to 870 cows and 744 heifers. 1633 female cattle of domestic origin were examined, most of them of dual-purpose type. Pregnancy was detected in 30/759 heifers and in 74/874 cows (an overall prevalence of 6.4%). The number of cattle in the last trimester of pregnancy was 16 to 26, depending on the evaluation scheme. We found no significant differences in percentages of pregnant cattle sent to slaughter for beef, dual-purpose and dairy breeds, although the latter group demonstrated the lowest percentage. Our results are comparable with those from previously conducted studies in other member states of the European Union. Measures to avoid sending pregnant cattle to slaughter should be implemented at farm-level.

## 1. Introduction

Both meat production and milk production are linked to the reproductive cycle of cattle. Calves are the essential basis of bovine production. The slaughter of pregnant cattle is not an exceptional event in Europe [1,2,3] and elsewhere, as evidenced from literature (Table 1). Various reasons may account for this, with accidents and disease on one hand and deficiencies in farm management on the other [2]. In non-dairy herds, pregnancy may be the result of natural mating when heifers, cows and bulls are kept together [2]. 

There is evidence that mammalian fetuses are perceptive and sentient at least from the last third of gestation [4], while other authors argue that these concerns are unfounded [5,6]. For animal welfare reasons, the transport of pregnant cattle is prohibited within the European Union if 90% of the gestation period has been exceeded [7]. Germany has banned the slaughter of cattle in the last third of gestation—except for medical indications—as of 1 September 2017 [8]. To date, there is no explicit European Union-wide ban on the slaughter of pregnant cattle after a certain gestation period. Consumption of meat from gravid animals could result in alimentary exposure of consumers to steroid hormones [9,10], since levels of estradiol-17ß in meat from pregnant cattle are up to ten times higher than in meat from non-pregnant cattle, especially from the end of the second trimester of pregnancy [1]. 

No data are available on the slaughter of pregnant cattle in Austria, and only pregnancy in the last 1/10th of gestation is recorded at meat inspection [23]. Thus, we tried to obtain an overview of the frequency of slaughter of pregnant cows and heifers at an Austrian slaughterhouse, with consideration of the stage of gestation, production type and seasonality. These data should allow comparison with reports from other countries, but also form the basis for further targeted activities involving the primary production sector [2].

## 2. Materials and Methods

### 2.1. Sampling Strategy

Sample size determination was based on an assumed prevalence of 2.5% (±1% accuracy; 95% confidence) of 3rd-trimester-pregnancy in the female cattle population admitted for slaughter. We selected a mid-size abattoir (facility was approved according to EU legislation), with an annual capacity of ca. 12,000 cows and 3600 heifers. The slaughtered cattle originate from ca. 2400 farms in the province of Carinthia, and from neighboring districts of East Tyrol and Styria [personal comm. of the owner of the abattoir], i.e., from subalpine to alpine regions. The average numbers of female cattle sent to slaughter are thus low, with ca. 5 cows and 1.5 heifers per farm and year, but can be explained by the low average herd size in Austria, i.e., 32 cattle per farm [24]. Number of samples was calculated with the tool provided by the Australian Bureau of Statistics (www.abs.gov.au, accessed on 25 September 2019) and amounted to 869 cows and 744 heifers. Heifers and cows were defined according to Regulation (EU) No 1308/2013 [25].

Sampling was performed over a one-year period (9 December 2019–16 November 2020). The examination took place one day every week, with the day in the week being randomly chosen. Per sampling day, an average of 16.5 heifers or 19 cows were examined.

An official veterinarian examined the uteri in the course of routine post-mortem inspection. The veterinarian also reported if gross pathological lesions had been observed during ante- and post-mortem inspection. Uteri were preserved until the end of the day of slaughter for a detailed examination. Then, the fetuses (if present) were extracted, head length (HL, mm), crown–rump length (CRL, mm), body weight (BW, g) were measured and degree of pubescence and development of the nipples or testicular descent were recorded.

Information on the age and breed of the dam were provided by the slaughterhouse operator. Due to data protection reasons, individual ear-tag numbers and information on the farm from which the animals originated were not made available for this study. However, the breed or crossbreed allowed us to conclude on the production type (beef/dairy/dual use) [26,27]. 

The number of animals examined corresponded to 0.43 and 0.60% of all heifers and cows slaughtered per year in Austria [28].

### 2.2. Assessment of the Age of the Fetuses

The age of the fetuses was determined by measuring crown–rump length (CRL) and comparing the CRL (i) to tabulated reference values for CRL and corresponding month of gestation [29,30] and (ii) calculating the month of pregnancy according to the equation given by Schnorr and Kressin [31]. Data were aggregated in 3-month intervals (trimester). 

Equation according to Schnorr and Kressin [31]: $$\left[\mathrm{month}\;\mathrm{of}\;\mathrm{pregnancy}=\left(\sqrt{\mathrm{CRL}\;\mathrm{in}\;\mathrm{cm}+1}\right)-1\right]$$

### 2.3. Statistics

The frequencies of pregnant animals in cows and heifers were compared by a 2 × 2 chi-square test. Likewise, the effect of three common schemes for determination of the age of the fetus on the classification of the pregnancy into trimesters was conducted by a 3 × 3 chi-square test. Level of significance was set to 0.05.

## 3. Results

### 3.1. Prevalence of Pregnant Cattle at Slaughter, According to Breed and Age Category

On average, 12,981 female cattle of Austrian origin were slaughtered at this abattoir in 2019 and 2020. This represented about 3.95% of all heifers and cows slaughtered in Austria [28].

Our study included 1633 cattle, i.e., 874 cows (53.5%) and 759 heifers (46.5%). The overall prevalence of pregnant cattle was 6.4% (95% C.I.: 5.2–7.6%; Table 2). Pregnancy was detected in 30 of the examined heifers (4%, 95% confidence interval (C.I.): 2.6–5.4%), as well as in 74 cows (8.5%; 95% C.I.: 6.7–10.4%), i.e., more frequently in cows than in heifers (chi square = 14.60; df = 1; *p* < 0.001).

For the prevalent cattle breed in Austria, i.e., Simmental (“Fleckvieh”, a dual-use breed), the overall frequency of pregnant cattle sent to slaughter was 6.5% (of *n* = 888; 95% C.I.: 4.9–8.1%) and for Simmental crosses, it was 2.7% (of *n* = 463; 95% C.I.: 95%: 1.2–4.2%). Additional information on more common breeds or crossbreeds is displayed in Table 2.

The numbers of cattle examined per month ranged from 33–104 for cows and 38–96 for heifers. Pregnant heifers were recorded in the period March to November (1.8–7.9%), whereas pregnant cows were found in all 12 months (3.0–7.2% in 9/12 months), with a peak in February, when 18/78 cows were pregnant. None of the 10 dairy cows, but 27.1% and 27.8% of dual-purpose and beef cows, respectively, were pregnant. A somewhat lower peak was observed in September and October, but with three pregnant dairy cattle out of 11 and 16/133 non-dairy cows. 

### 3.2. Gestational Stage of Pregnant Cattle per Trimester and Frequency of Cattle Slaughtered in the Last Month of Pregancy 

The numbers of pregnant cattle according to age category (heifer/cow) and trimester are displayed in Figure 1. 

For the prevalent breeds, i.e., pure Simmental and Simmental crosses, data are displayed separately; other breeds were aggregated into one category. 

The different age classification schemes yielded slightly different pregnancy frequencies for each trimester (Figure 1); however, this was not statistically significant (cows: chi square = 2.78; df = 4; *p* = 0.59; heifers: chi square = 0.92; df = 4; *p* = 0.92). 

Based on fetal age assessment by Schnorr and Kressin [31], a total of 16 cattle in the last trimester of gestation and three cattle in the last month of gestation were slaughtered, which corresponds to 0.98 and 0.18% of the total tested cattle, respectively.

### 3.3. Prevalence of Pregnant Cattle at Slaughter, According to Production Type and Age Category

We compared the frequency of pregnant cattle and heifers according to the production type (beef, dairy or dual-purpose)—(Table 3), age of the dam and of the fetus (according to [31]).

For heifers, pregnancy was detected in 5.2% of dual-purpose breeds (of *n* = 309; 95% C.I.: 2.7–7.7%), in 3.1% of beef breeds or crosses (*n* = 415; 95% C.I.: 1.5–4.8%) and in 2.9% of dairy (*n* = 35; 95% C.I.: 2.7–8.4%).

For cows, pregnancy was detected in 7.9% of dual-purpose breeds (of *n* = 611; 95% C.I.: 5.7–10.0%), in 14.4% of beef breeds or crosses (*n* = 132; 95% C.I.: 8.4–20.4%) and in 5.3% of dairy (*n* = 131; 95% C.I.: 1.5–9.2%). Numbers for dairy breeds were lowest, and those for beef breeds highest. Since the confidence intervals overlapped, no statistically significant difference could be established.

### 3.4. Characterisation of the Fetuses

Morphometric characteristics for the fetuses detected in the 104 gravid cattle are given in Table 4. Stage of pregnancy was reported in trimesters. Estimation of the age of the fetus was performed according to Schnorr and Kressin [31]. Multiple pregnancies were detected in six cattle (five twin and one triplet pregnancy). Since fetal length and weight differed considerable between twins or triplets, classification of month of gestation of the cow was based on the largest fetus.

## 4. Discussion

### 4.1. Methods for Estimation of the Age of Bovine Fetuses

The determination or calculation of the age of the fetuses using the three different methods led to somewhat different results, with Habermehl [29] estimating the age of the fetuses to be the highest and Schnorr and Kressin [31] the lowest; the age determination according to Richter et al. [30] lay between the two schemes. According to Richardson et al. [32] the vertex–anus length follows an approximately linear function with a slight kink after the 150th day of gestation. Similarly, the three schemes we applied approximately follow a linear progression. A number of other measurements, (e.g., body weight) and also qualitative indicators are in use for estimation of the age of fetuses and the combination of several factors allows to calculate the age not by month, but by day, (e.g., Nielsen et al. [2]). Although such data were recorded in this study, we decided to base our age assessment solely on the crown–rump length and used three established classification schemes [29,30,31]. The rationale was that these schemes were developed with a focus on Germany and neighboring countries, with similar cattle (cross)breeds as in Austria. This is of particular relevance when differences in fetus dimensions between breeds are assumed. Furthermore, the age estimation according to Schnorr and Kressin [31] allowed a direct comparison of results with those from a German study [Riehn K, personal communication]. However, this scheme was obviously the most conservative one. Other age assessment schemes yield higher numbers, but it is conceivable that authorities would rely on the first grading scheme, since it represents the consensus of the three grading schemes and would most likely be used when a case goes to court.

### 4.2. Prevalence of Pregnant Animals among Cows and Heifers Sent to Slaughter

The overall prevalence of pregnant cattle was 6.4%, with 8.5% in cows and 4.0% in heifers. Similar findings were reported in other studies from Switzerland, Germany, Luxembourg, Belgium and Italy (Table 1). A prevalence of >20% has been reported in European and non-European countries, and the reasons for such a high prevalence has been explored and discussed in several works, (e.g., [2]). The percentage of fetuses in the last trimester of pregnancy we reported (14.4% or 16/111) is somewhat lower than reported from other countries [1,9]. However, due to differences in fetal age assessment schemes and differences in the use of cattle (dairy or non-dairy, [2]) comparison of these data should be done with caution.

We observed a lower percentage of pregnant cattle among the dairy cows compared to dual-use and beef breeds. This might be indicative of a more intensive health management of dairy herds, or due to common pasture of females and males in beef- or dual-purpose-herds [2]. However, the reported percentages were not statistically significantly different. The peak in the frequency of pregnant cows in February, and the prevalence of non-dairy cows amongst the pregnant ones could be explained as the result of uncontrolled mating during the alpine pasturing in the previous summer. The somewhat lower peak in September and October included also pregnant dairy cows. We refrained from further interpretation of the data since essential information on the farms from which the animals originated was not available to us. 

In Germany, the scheme according to Schnorr and Kressin [31] was or is applied [Riehn K, personal communication]. According to this scheme, 16 cattle were in the last trimester of gestation, and thus according to German law [8] would not have been accepted for slaughter. 

According to Schnorr and Kressin [31] three cows were in the ninth month; according to Habermehl [29] it would be 11 and according to Richter et al., [30] seven. If a gravidity in the ninth month was determined, further signs of maturity such as presence of hair would have to be included to verify whether the last tenth of gravidity had to be assumed and thus a violation of Council Regulation (EC) No 1/2005 [7] was given or not.

### 4.3. Consumer Protection Concerns about Meat from Pregnant Cattle

Kushinsky [15] reported that steroid hormone tissue levels in pregnant cows are several times higher than in hormone-supplemented beef cattle in the United States. Levels of progesterone and estradiol-17ß measured by Riehn et al. [1] in the muscle of gravid cattle are on average higher than those reported by Kushinsky [15] at all stages of gestation. Compared to non-gravid animals, levels of estradiol-17ß were found to be tenfold higher in some cases. Estradiol-17ß is considered the most potent natural estrogen. An increase in steroid hormones in the muscle and fat of gravid animals was particularly evident from the end of the second trimester of gestation.

However, the analytical procedures used to quantify steroid hormones still have some shortcomings. The high individual variability appears to be problematic when evaluating results. Additionally, the data situation regarding the hormone content in the tissues of pregnant animals and the prevalence of pregnant slaughtered animals is to be considered as insufficient.

The SCVPH [33,34,35] explicitly points out the mutagenic and genotoxic potential of estradiol-17ß. Additionally, no ADI dose could be established to date. It cannot be excluded that steroid hormones from the meat of gravid animals constitute a hazard to consumers due to alimentary exposure to in principle.

Since infectious diseases are amongst the more common reasons for sending pregnant cattle to slaughter [2], it could be expected that animal tissues might contain residues from veterinary drugs. However, information on the use of pharmaceuticals must be given to the slaughterhouse as part of the food chain information.

### 4.4. Sensitivity of the Fetuses

To date, it is controversial whether and from which developmental stage fetuses can consciously feel pain and stress. Previous studies, e.g., Mellor et al. [5,6] assume that fetuses lack such abilities. This may need to be revised in light of new scientific evidence. Bellieni and Buonocore [4] report that fetuses are able to feel distress and pain from the second half of gestation. The Experimental Animals Directive 2010/63/EU amending Directive 86/609/EEC [36] on the protection of animals used for scientific purposes already takes into account the new research findings and states in recital nine that there is scientific evidence showing that fetal forms of mammals are at increased risk of experiencing pain, suffering and distress in the last third of their developmental period. Overall, it cannot be ruled out that fetuses experience pain, distress and other forms of suffering, and the slaughter of pregnant cattle at an advanced stage of gestation is considered an animal welfare issue.

## 5. Conclusions

We could demonstrate that the slaughter of pregnant cattle occurs in Austria. Out of 1633 examined female cattle, 104 were pregnant, and three of them were in the last month of gestation. There was an indication that this occurs more frequently in non-dairy cattle, and a higher percentage of pregnant non-dairy cows slaughtered in February might be indicative of uncontrolled mating during (sub-)alpine pasturing in summer. The lack of information on structure and management of the farms from which the pregnant cattle originated is a clear limitation of the study. Thus, further studies will have to include such data in order to explore why pregnant cattle are sent to slaughter and how this can be remedied. Such issues have already been addressed in other countries. The authors hope that the results of this pilot study will help to initiate similar activities in Austria.

## Figures and Tables

**Figure 1 animals-11-02474-f001:**
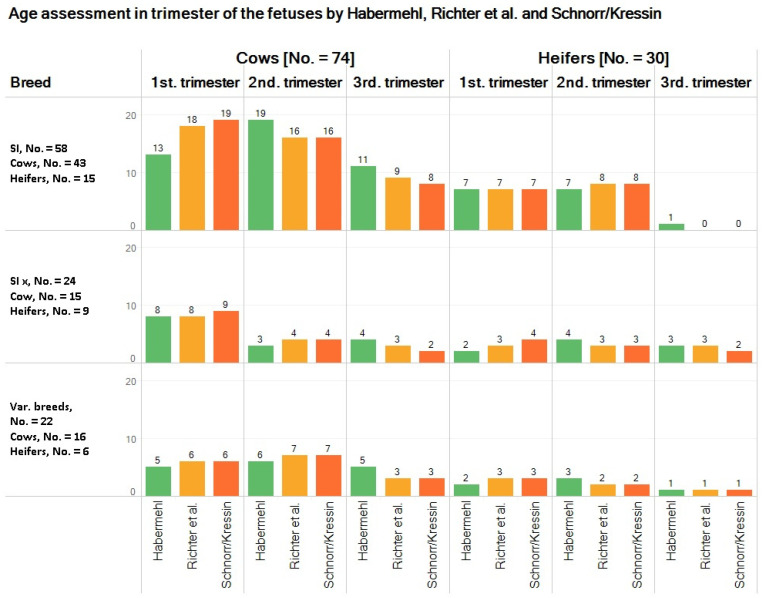
Gestational stage of cows and heifers, according to breed and to classification scheme. Note: SI = Simmental; SI x = Simmental-Crossbreed.

**Table 1 animals-11-02474-t001:** Literature data on the prevalence of pregnant animals amongst cattle submitted to slaughter.

Country	Prevalence, Percentage; Gravid/Total	Stage of Gestation (Month or Trimester = X/3) and % of Animals	Method for Estimation of Stage of Gestation	Reference
Austria	6.4%; 104/1633	1/3: 45.2%, 2/3: 39.4%, 3/3: 15.4%	Schnorr and Kressin	This study
Belgium	10.1%; 97/965		Schnorr and Kressin	Di Nicolo 2006 [9]
Denmark	23%; 187/814	1/3: 16%, 2/3: 5%, 3/3: 2%	Krog et al.	Nielsen et al., 2019 [2]
Germany	Up to 10%, mean 4.3% of cows and heifers			Lücker et al., 2003 [10]
Germany	4.9%; 77/1556	mostly in 5th month; 2/3: 38%, 3/3: 62% in 3/3	Schnorr and Kressin	Di Nicolo 2006 [9]
Germany	Up to 15%, mean 9.6%, median 7.1% of cows and heifers	2/3 and 3/3: 90%		Riehn et al., 2011 [1]
Germany	8.2%; 561/7005	1/3: 32.4%, 2/3: 46.7%, 3/3: 20.9%	Schnorr and Kressin	Riehn et al., 2019 [11]
Italy	4.5%; 138/3071	3/3: 15%	Schnorr and Kressin	Di Nicolo 2006 [9]
Luxembourg	5.3%; 164/3099	3/3: 36%	Schnorr and Kressin	Di Nicolo 2006 [9]
Switzerland	5.67%	>5 months	Habermehl, Richter et al.	EDI BLV 2014 [12]
United Kingdom	23.4%; 1885/8071	1/3: 22.1%, 2/3: 50.9%; 3/3: 25.0%	Crown–rump length	Al-Dahash and David 1977 [13]
United Kingdom	23.5%; 588/2502	3/3: 26.9%	Crown–rump length	Singleton and Dobson 1995 [3]
USA	25.5%; 255/1000, cows and heifers			Perkins et al., 1954 [14]
USA	approx. 5%			Kushinsky 1983 [15]
Canada		1/3: 13.1%, 2/3: 62.6%, 3/3: 24.3%		Herenda 1987 [16]
Australia	63% cows and heifers; 4721/7495			Ladds et al., 1975 [17]
Cameroon	16.61%; 5778/34,780	1/3: 45%, 2/3: 34.5%, 3/3: 20.5%		Tchoumboue 1984 [18]
Nigeria	9.77%; 5654/57,891	1/3: 42.3%, 2/3: 38.3%, 3/3: 19.4%		Wosu 1988 [19]
Nigeria	50.9%	1/3: 26%, 2/3: 67%, 3/3: 7%		Ojo et al., 1978 [20]
Tanzania	29.1%; 655/2256	1/3: 25.8%, 2/3: 42.7%, 3/3: 31.6%		Swai et al., 2015 [21]
Pakistan	8.6%; 28/325			Khan and Khan 1989 [22]

Note: empty cells = no data reported.

**Table 2 animals-11-02474-t002:** Overview on animals included in this study, and frequency of pregnant female cattle submitted to slaughter, by age category and breed.

	Total		Heifers		Cows	
Breed/Crossbreed	*n*	Pregnant, No. (%)	*n*	Pregnant, No. (%)	*n*	Pregnant, No. (%)
Simmental ^d^	888	58 (6.5)	298	15 (5.0)	590	43 (7.3)
Simmental crossbreed ^b^	463	24 (5.2)	350	9 (2.6)	113	15 (13.3)
Holstein Friesian ^m^	78	4 (5.1)	7	0 (0.0)	71	4 (5.6)
Charolais ^b^	36	3 (8.3)	18	1 (5.5)	18	2 (11.1)
Carinthian Blonde ^b^	28	5 (17.9)	11	2 (18.2)	17	3 (17.6)
Brown Swiss ^m^	25	1 (4.0)	14	1 (7.1)	11	0 (0,0)
Pinzgauer ^d^	18	3 (16.7)	6	0 (0.0)	12	3 (25.0)
Others [45 (cross)breeds]	97	6 (6.2)	55	2 (3.6)	42	4 (9.5)
All breeds/crossbreeds	1633	104 (6.4)	759	30 (4.0)	874	74 (8.5)

Note: ^b^ = beef, ^d^ = dual-purpose, ^m^ = dairy (cross-) breed; based on [26,27].

**Table 3 animals-11-02474-t003:** Number (No.) and prevalence of pregnant cattle sent to slaughter, stratified by production type, parity (age category) and stage of gestation.

		Trimester of Gestation							
	All animals	Overall		1/3		2/3		3/3	
		No.	%	No.	%	No.	%	No.	%
All animals	1633	104	6.4	48	2.9	40	2.4	16	1.0
Dairy									
-all animals	166	8	4.8	4	2.4	2	1.2	2	1.2
-heifers	35	1	2.9	1	2.9	0	0.0	0	0.0
-cows	131	7	5.3	3	2.3	2	1.5	2	1.5
Beef									
-all animals	547	32	5.9	14	2.6	12	2.2	6	1.1
-heifers	415	13	3.1	5	1.2	5	1.2	3	0.7
-cows	132	19	14.4	9	6.8	7	5.3	3	2.3
Dual-purpose									
-all animals	920	64	7.0	30	3.3	26	2.8	8	0.9
-heifers	309	16	5.2	8	2.6	8	2.6	0	0.0
-cows	611	48	7.9	22	3.6	18	2.9	8	1.3

**Table 4 animals-11-02474-t004:** Morphometric characteristics of fetuses, according to age category of cattle and trimester of pregnancy (estimated according to the equation given by Schnorr and Kressin [31]).

Characteristic	Heifers (*n* = 31) *: M ± SD			Cows (*n* = 80) *: M ± SD		
	1st	2nd	3rd trimester	1st	2nd	3rd trimester
CRL mm	86 ± 54	357 ± 103	653 ± 118	89 ± 54	334 ± 108	781 ± 121
Head Length mm	32 ± 23	126 ± 26	203 ± 28	35 ± 24	124 ± 31	224 ± 33
Body Mass g	78 ± 118	3248 ± 2613	15,700 ± 6437	79 ± 95	2756 ± 2589	27,286 ± 13,940

Note: M = mean, SD = standard deviation, CRL = crown–rump length; * number of fetuses.

## Data Availability

Datasets analyzed in this study are publicly available. This data can be found here: https://phaidra.vetmeduni.ac.at (accessed on 23 August 2021).

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
