# Peer review of "Slaughter of Pregnant Cattle at an Austrian Abattoir: Prevalence and Gestational Age"

_animals, 2021, doi:10.3390/ani11082474_

Round 1
Reviewer 1 Report
General comments:
Despite data collection has being carried out in Austria, this manuscript addresses an issue of general interest, since the situation tackled on it also occurs in other countries. The research design is very simple, but it approach a subject and brings outcomes that lead us to pay more attention to an important practice that is not usually taken into consideration, the slaughter of pregnant cows, which potentially impoverishes the welfare of cow and fetus in late gestation, as well as brings health risk to humans.
Suggestions
L69 - Exclude “.” after “2020)."
L113 - Exclude “- Table 3”
L120-121 (Table 3) - This data seems to me not relevant, I suggest excluding Table 3.
L161 - Replace “Table 6” for “Table 3”.
L179 - Replace “6” for “3”.
The content of the third column of [former] Table 6 is very confusing, it is necessary to improve it.
Author Response
Response to the comments of reviewer 1:
Reviewer: Despite data collection has being carried out in Austria, this manuscript addresses an issue of general interest, since the situation tackled on it also occurs in other countries. The research design is very simple, but it approach a subject and brings outcomes that lead us to pay more attention to an important practice that is not usually taken into consideration, the slaughter of pregnant cows, which potentially impoverishes the welfare of cow and fetus in late gestation, as well as brings health risk to humans.
Authors: Thank you very much for your comments and for the confirmation that the topic is of general interest. We are aware that a report from one country only and with a simple study design has some limitations, but we agree that such reports may rise general awareness on practices conflicting with animal welfare.
Reviewer: L69 - Exclude “.” after “2020)."
Authors: corrected
Reviewer: L113 - Exclude “- Table 3”
Authors: corrected
Reviewer: L120-121 (Table 3) - This data seems to me not relevant, I suggest excluding Table 3.
Authors: table has been deleted.
Reviewer: L161 - Replace “Table 6” for “Table 3”; L179 - Replace “6” for “3”.
Authors: changes done / corrected.
Reviewer: The content of the third column of [former] Table 6 is very confusing, it is necessary to improve it.
Authors: we have split the information in the column in several coumns, and explained better that 1/3, 2/3.. refer to trimesters of pregnancy.
Reviewer 2 Report
An interesting study on the frequency, and stage, of pregnancy in cattle at slaughter. The table in the discussion should go into the introduction to give context of the condition.
There also seems to be a missed opportunity to not statistically evaluate the breeds, which seem to differ and this could possibly be explained by the level of management of said animals. I expect more close involvement of the vet in fertility with pure dairy (HF) than with beef, and the dual purpose breeds possibly in between. Also, was there seasonal variation on the pregnancies and stage found? This is not explored, nor presented, but could target the proposed intervention of more checks and controls. Overall, the statistical evaluation needs more exploring and more clearly presenting.
There is a level of interpretation in the results, which should be avoided, obviously. There is an obvious misuse of the word in line 129.
Author Response
Response to the comments of reviewer 2:
Reviewer: An interesting study on the frequency, and stage, of pregnancy in cattle at slaughter. The table in the discussion should go into the introduction to give context of the condition.
Authors: thank you very much for this interest in the topic. We have now moved the former table 6 in the introduction, as suggested. The table as such underwent some revisions as suggested by reviewer 1.
Reviewer: There also seems to be a missed opportunity to not statistically evaluate the breeds, which seem to differ and this could possibly be explained by the level of management of said animals. I expect more close involvement of the vet in fertility with pure dairy (HF) than with beef, and the dual purpose breeds possibly in between.
Authors: thank you very much for this valuable suggestion. We assigned the breeds/crossbreeds to their typical use and generated a new table displaying number and frequency of pregnant cattle according to age (heifer/cow), use (dairy, beef, dual), similar to that in the paper of Nielsen et al. (2019). In fact, the highest percentages of pregnant cattle sent to slaughter was in the "beef" use group, somewhat lower in the "dual purpose" group, and lowest in the "dairy" group. Hower, the confidence intervals overlapped, so it was not statistically significant. This information is now included in the manuscript.
Reviewer: Also, was there seasonal variation on the pregnancies and stage found? This is not explored, nor presented, but could target the proposed intervention of more checks and controls.
Authors: we checked for percentages of pregant cattle by month of slaughter and found a higher percentage in cows slaughtered in February (>20%), followed by September/October (>10% each month). Since the majority of cows were dual purpose, the numbers of pregnant cows were also higher in this group. The "peak" in February was exclusively attributable to dual purpose and beef breeds, and may be due to uncontrolled mating when cattle had been pastured on the (sub)alpine meadows the previous summer. This is now presented in the manuscript. We feel that for a more in-depth data analysis is hampered by the fact that we have no information on the farms from which the animals originated and only a very general information on the region of origin (Carinthia and adjacent districts of Styria and East Tyrol). This is one of the limitations we had to accept before starting the study.
Reviewer: Overall, the statistical evaluation needs more exploring and more clearly presenting.
Authors: We have restructured and amended the evaluation of data, and hope that it is now both more in-depth and easier to read.
Reviewer: There is a level of interpretation in the results, which should be avoided, obviously. There is an obvious misuse of the word in line 129.
Authors: We deleted this sentence/statement.
Reviewer 3 Report
This manuscript contributes an important survey to our understanding of conditions in which bovine females are sent to slaughter. Generally, sending pregnant females to slaughter results from poor record-keeping, poor communication or ill-informed decisions on the farm. For dual-purpose production systems, slaughtering otherwise healthy pregnant females has a significant economic impact.
This manuscript adds to information contributed by authors from other countries. What the manuscript lacks, but it may be quite difficult to obtain, is a demographic profile of farms whence pregnant females originated. This information would be helpful in devising interventions and educational approaches for farms that are likely to contribute to this pool. Additionally, although not likely possible from this research, the authors endeavor to describe effects of potentially greater estrogenic concentrations in muscle and organ meat products derived from pregnant females. Yet, a greater concern would be concentrations of antibiotics or other therapeutic agents still active in muscle and organ meat products derived from pregnant females.
Line Comment
15 It would be helpful to know what proportion of total yearly harvest the number of cattle examined (1,633) comprise
25 Although requiring access to on-farm records, identifying the demographics associated with farms where pregnant females were sourced would be extremely helpful. This information could be used to recommend interventions and devise educational approaches for farms for which this situation occurs more frequently.
77 Though this reviewer understands this situation, it is important to determine and present descriptive observations regarding the population of cattle and farms the abattoir serves
215 “cattle are sent to slaughter”
Author Response
Response to the comments from reviewer 3:
Reviewer: This manuscript contributes an important survey to our understanding of conditions in which bovine females are sent to slaughter. Generally, sending pregnant females to slaughter results from poor record-keeping, poor communication or ill-informed decisions on the farm. For dual-purpose production systems, slaughtering otherwise healthy pregnant females has a significant economic impact.
This manuscript adds to information contributed by authors from other countries.
Reviewer: What the manuscript lacks, but it may be quite difficult to obtain, is a demographic profile of farms whence pregnant females originated. This information would be helpful in devising interventions and educational approaches for farms that are likely to contribute to this pool.
Authors: we agree that such information is extremely relevant, but we can just give a general description on numbers of farm sending cattle to the abattoir, and on the average herd sizes in this region (small herds). However, we assigned the breeds/crosses to their typical use and created a table which shows a tendency that it is less likely that pregnant dairy cows are sent to slaughter.
Reviewer: Additionally, although not likely possible from this research, the authors endeavor to describe effects of potentially greater estrogenic concentrations in muscle and organ meat products derived from pregnant females. Yet, a greater concern would be concentrations of antibiotics or other therapeutic agents still active in muscle and organ meat products derived from pregnant females.
Author: thank you very much for this comment. We discussed to steroid hormones issue, since it is presented in many studies on pregnant cattle and to give a motivation beyond animal welfare and ethics. As regards residues from drugs, we refrained from discussing this issue, since each cattle sent to slaughter is accompanied by a statement that no drugs have been applied and there are no withdrawal times to consider. And we did not want to put this in question.
Reviewer: Line 15: It would be helpful to know what proportion of total yearly harvest the number of cattle examined (1,633) comprise
Authors: we have inserted the annual number of cattle slaughtered in this particular abattoir and in whole Austria.
Reviewer: Line 25: Although requiring access to on-farm records, identifying the demographics associated with farms where pregnant females were sourced would be extremely helpful. This information could be used to recommend interventions and devise educational approaches for farms for which this situation occurs more frequently.
Authors: We are aware that such information is extremely important, but we could not provide it. We have now added a sentence highlighting that the lack of such information is a clear limitation of our study. However, we hope that this initial research will facilitate further more in-depth works.
Reviewer: line 77: Though this reviewer understands this situation, it is important to determine and present descriptive observations regarding the population of cattle and farms the abattoir serves
Authors: we fully agree with this comment. All additional information we could retrieve is now included in the revision. As written in the reply to the previous comment, we acknowledge that this lack of data is a limitation of our study, but hope that subsequent research will be able to use a broader set of data. We had to make some compromises before starting the study.
Reviewer: line 215 “cattle are sent to slaughter”
Authors: corrected.
English language: we had the revised manuscript checked for grammar and typos.
Round 2
Reviewer 1 Report
The changes made in the manuscript improved it substantially, and are in line my suggestions. It is now ready to be published.
Reviewer 2 Report
I am happy with the changes.